# Characterization and Transcriptional Regulation of the 2-Ketogluconate Utilization Operon in *Pseudomonas plecoglossicida*

**DOI:** 10.3390/microorganisms12122530

**Published:** 2024-12-08

**Authors:** Lei Sun, Wenqi Yang, Lulu Li, Daming Wang, Xinyi Zan, Fengjie Cui, Xianghui Qi, Ling Sun, Wenjing Sun

**Affiliations:** School of Food and Biological Engineering, Jiangsu University, Zhenjiang 212013, China; sunlei@ujs.edu.cn (L.S.); 2222318065@stmail.ujs.edu.cn (W.Y.); 15380664161@163.com (L.L.); wdm9987@163.com (D.W.); zxy19880920@163.com (X.Z.); fengjiecui@163.com (F.C.); qxh@ujs.edu.cn (X.Q.); sunling090909@163.com (L.S.)

**Keywords:** *Pseudomonas plecoglossicida*, 2-keto-d-gluconate (2KGA), 2-ketogluconate utilization operon (*kgu* operon), transcription regulator PtxS, transcriptional regulation mechanism

## Abstract

*Pseudomonas plecoglossicida* JUIM01 is an industrial 2-keto-d-gluconate (2KGA)-producing strain. However, its regulation mechanism of 2KGA metabolism remains to be clarified. Among other reported *Pseudomonas* species, the 2-ketogluconate utilization operon (*kgu* operon) plays key roles in 2KGA catabolism. In this study, the structural genes of the *kgu* operon and its promoter in *P. plecoglossicida* JUIM01 were identified using reverse transcription PCR and *lacZ* reporter gene fusion. The results showed the *kgu* operon in *P. plecoglossicida* was composed of four structural genes: *kguE*, *kguK*, *kguT*, and *kguD*. The *ptxS* gene located upstream of *kguE* was excluded from the *kgu* operon. Then, the knockout and corresponding complementation strains of *kguE*, *kguK*, *kguT*, and *kguD* were constructed, respectively, to prove the *kgu* operon was involved in 2KGA catabolism of *P. plecoglossicida*. The knockout stains, especially JUIM01Δ*kguE*, showed potential as industrial production strains for 2KGA. Moreover, the transcriptional regulation mechanism of PtxS on the *kgu* operon was elucidated using multiple methods. In *P. plecoglossicida*, the LacI-family transcription regulator PtxS could recognize a 14 bp palindrome (5′-TGAAACCGGTTTCA-3′) within the promoter region of the *kgu* operon and specifically bind to a 26 bp region where the palindrome was located. As the binding sites overlapped with the transcription start site of the *kgu* operon, the binding of PtxS possibly hindered the binding of RNA polymerase, thus repressing the transcription of the *kgu* operon and further regulating 2KGA catabolism. 2KGA bound to PtxS as an effector to dissociate it from the *kgu* operon promoter region, so as to relieve the transcription repression. The results will provide strategies for improving the product accumulation in 2KGA industrial production and theoretical bases for the construction of a *Pseudomonas* chassis.

## 1. Introduction

2-Keto-d-gluconate (2KGA), an intermediate of glucose metabolism in several oxidative bacteria, can be used as a renewable raw material for regioselective or stereoselective chemical reactions in the synthesis of heterocyclic and other functional compounds [1]. The major application of 2KGA is as the precursor to synthesize d-erythorbate (d-isoascorbate, isovitamin C) and its salts, which are widely used food antioxidants [2,3,4]. Currently, 2KGA is produced industrially by fermentation, and *Pseudomonas plecoglossicida* JUIM01 is an industrial 2KGA-producing strain in China [5,6,7,8].

The genes directly related to 2KGA metabolism in *Pseudomonas* mainly include *gcd* (encoding glucose dehydrogenase); the glucose acid dehydrogenase operon composed of *gndS*, *gndL*, and *gndC* (encoding three subunits of gluconate dehydrogenase); *kguE* (encoding a putative epimerase); *kguK* (encoding 2-ketogluconate kinase); *kguT* (encoding 2-ketogluconate transporter); and *kguD* (encoding 2-keto-6-phosphogluconate reductase) [5,9,10,11,12,13]. Among them, *gcd*, *gndS*, *gndL*, and *gndC* play roles in 2KGA synthesis [14,15], while *kguE*, *kguK*, *kguT*, and *kguD* are related to its catabolism (Figure 1A). In *Pseudomonas putida* KT2440, the latter four genes constitute the 2-ketogluconate utilization operon (*kgu* operon) [11]. However, there is debate about the composition of the *kgu* operon in *Pseudomonas aeruginosa* PAO1: Swanson et al. (2000) suggest that these four genes form the *kgu* operon along with the upstream *ptxS* [16], while Daddaoua et al. (2012) maintain that *ptxS* is independent of the *kgu* operon [12]. We have cloned a DNA fragment containing the *kgu* operon from the *P. plecoglossicida* JUIM01 genome and conducted bioinformatics analysis. The results indicate that *ptxS* is unlikely a part of the *kgu* operon in *P. plecoglossicida* [17].

PtxS is a LacI-family transcription factor ubiquitous in *Pseudomonas* species [12]. In *P. putida* and *P. aeruginosa*, PtxS functions as a transcriptional repressor that specifically recognizes and binds to a 14 bp palindrome (5′-TGAAACCGGTTTCA-3′), thereby globally regulating the metabolism of 2KGA, and 2KGA generates feedback control on PtxS as an effector [11,12,16,18,19,20,21]. However, the metabolic regulation mechanism of 2KGA in other *Pseudomonas* species, especially in industrial 2KGA-producing strains, remains to be elucidated. We previously demonstrated that the PtxS of *P. plecoglossicida* can also specifically recognize and bind to the 14 bp palindrome (5′-TGAAACCGGTTTCA-3′) and 2KGA is the effector of PtxS [22]. Further analysis of the *P. plecoglossicida* JUIM01 genome revealed that the upstream promoter region of *kguE* contains the aforementioned 14 bp palindrome [17].

Based on the above studies, we made a prediction that the *kgu* operon in *P. plecoglossicida* JUIM01 was composed of *kguE*, *kguK*, *kguT*, and *kguD*, directly related to 2KGA catabolism, and was regulated by the LacI-family transcription factor PtxS. Therefore, in the present study, the *kgu* operon in *P. plecoglossicida* JUIM01 was identified using reverse transcription PCR and promoter fusion. Whereafter, the role of the *kgu* operon in the 2KGA degradation metabolism of *P. plecoglossicida* was verified by the knockout and complementation of each structure gene. The transcriptional regulation mechanism of PtxS on the *kgu* operon was further analyzed using multiple methods. The results would provide strategies for improving the product accumulation in 2KGA industrial production and theoretical bases for the construction of a *Pseudomonas* chassis.

## 2. Materials and Methods

### 2.1. Strains, Plasmids, and Primers

The strains and plasmids used are shown in Table 1.

The knockout of *kguE*, *kguD,* and *ptxS*, as well as the complementation of *kguE* and *kguD* in *P. plecoglossicida* JUIM01, were carried out using reported methods [5,23], with the corresponding primers changed (Appendix A). The knockout and complemented strains of *kguT* and *kguK* have been previously constructed [5,23].

Using the JUIM01 genome as the template and K1/K2 as the primers (Appendix A and Figure 1B), a *ptxS-kguE* transgene fragment with 20 bp vector homologous arms at both ends was amplified. The fragment was seamlessly linked with a linearized pME6522 vector using the ClonExpress^®^ II One Step Cloning Kit (Nanjing Vazyme Biotech Co., Ltd., Nanjing, China) to construct the recombinant plasmid pME6522-*kgu*. pME6522-*kgu* was then electroporated into JUIM01 and JUIM01Δ*ptxS* competent cells to construct recombinants JUIM01/pME6522-*kgu* and JUIM01Δ*ptxS*/pME6522-*kgu*, respectively. JUIM01/pME6522 was also constructed as the negative control.

The primers used are shown in Appendix A and Figure 1B and Appendix A, which were synthesized by Sangon Biotech Co., Ltd. (Shanghai, China).

### 2.2. lacZ Reporter Gene Fusions

The recombinants JUIM01/pME6522, JUIM01/pME6522-*kgu*, and JUIM01Δ*ptxS*/pME6522-*kgu* were separately spread on LB plates (containing 10 g/L NaCl, 10 g/L tryptone, 5 g/L yeast extract, and 15 g/L agar) with 10 μg/mL tetracycline and incubated overnight at 30 °C. Single colonies were picked from the plates and inoculated into 5 mL of LB liquid media (containing 10 g/L NaCl, 10 g/L tryptone, 5 g/L yeast extract, and 20 g/L glucose) with 10 μg/mL tetracycline and incubated at 30 °C and 180 rpm for 12 and 24 h. The OD_650_ and *β*-galactosidase activities of the culture were then measured using a spectrophotometer and the *β*-Galactosidase Enzyme Assay System with Reporter Lysis Buffer (Promega, Madison, WI, USA).

### 2.3. Comparison of 2KGA Utilization Abilities

The activated *P. plecoglossicida* JUIM01 and its derived gene-deletion strains were resuspended in 1 mL of sterile water, and then inoculated into 500 mL shake flasks containing 50 mL of seed media (20.0 g/L calcium 2-ketogluconate, 10.0 g/L corn steep liquor, 2.0 g/L urea, 2.0 g/L KH_2_PO_4_ and 0.5 g/L MgSO_4_·7H_2_O, pH 7.0) at 30 °C and 265 rpm on a rotary shaker for 24 h. Cell growth and 2KGA concentration were measured using reported methods [6,23].

### 2.4. 2KGA Fermentation

The activated cells were inoculated into the 50 mL of seed media (20.0 g/L glucose, 5.0 g/L corn syrup powder, 2.0 g/L urea, 2.0 g/L KH_2_PO_4_, 0.5 g/L MgSO_4_·7H_2_O and 5.0 g/L CaCO_3_, pH 7.0) in 500 mL shake flasks and cultivated at 33 °C and 265 rpm for at least 28 h.

### 2.5. Primer Extension

The total RNA of *P. plecoglossicida* JUIM01 was extracted using the RNAPrep Pure Cell/Bacteria Kit (TIANGEN Biotech Co., Ltd., Beijing, China). The primer *kguE*-PE(FAM) was designed and synthesized for the *kgu* operon and its upstream sequence, with a FAM label at the 5′ end (Appendix A and Appendix A). Primer extension was performed based on a reported protocol [25].

### 2.6. Electrophoretic Mobility Shift Assay (EMSA)

The DNA probe JUIM01-*kgu*(FAM) (labeled with FAM at its 5′ end) containing the *kgu* operon promoter region sequence was synthesized by Sangon Biotech Co., Ltd. (Shanghai, China). The EMSA was performed based on a reported protocol [26]. First, the amount of the probe JUIM01-*kgu*(FAM) remained 50 ng in the reaction system, and the amount of PtxS protein was 0 for the control lane and 0.5 μg for the other lanes to investigate the specific binding of PtxS to the *kgu* operon promoter region. The amount of 2KGA then gradually increased from 0 to 50 mM, while the amounts of both JUIM01-*kgu*(FAM) and PtxS remained constant, to investigate the effect of the effector 2KGA on the binding of PtxS to the *kgu* operon promoter region. The heterologous expression and purification of PtxS were previously reported [22].

### 2.7. DNase I Footprinting

The 400 ng JUIM01-*kgu*(FAM) probe was used to bind to different amounts (0, 0.9, and 2.2 μg) of PtxS at 25 °C for 30 min, and then was digested with 0.015 U DNase I (Promega, Madison, WI, USA) at 37 °C for 1 min. The digested sample was extracted with a phenol–chloroform–isoamyl alcohol mixture (25:24:1) to remove protein, and the supernatant was added to 2.5-fold the volume of absolute ethanol and 1/10 volume of 3 M sodium acetate. The sample was placed at −20 °C for 1 h to precipitate DNA and centrifuged to remove the supernatant. The precipitate was washed with 1 mL of 75% ethanol, centrifuged, and dried. The resulting DNA was dissolved in Mili-Q ultrapure water and detected by an ABI 3130xl Genetic Analyzer (Foster City, CA, USA), using GeneScan™ 500 LIZ^®^ Size Standard (ABI, Foster City, CA, USA) as the ladder. The Peak Scanner software v1.0 (Applied Biosystems) was used to output the fluorescence peak graphs and base sequences based on the sequencing results.

### 2.8. Real-Time Fluorescence Quantitative PCR (RT-qPCR)

The exponential-phase cells of *P. plecoglossicida* JUIM01 and JUIM01Δ*ptxS* were collected to extract their RNAs using the UNIQ-10 column Trizol total RNA extraction kit (Sangon Biotech Co., Ltd., Shanghai, China). The RNAs were reversely transcribed using the FastQuant RT Kit (With gDNase) (TIANGEN Biotech Co., Ltd., Beijing, China) to prepare corresponding cDNAs. Taking the cDNAs as the templates, the upstream and downstream primers were designed according to the conserved regions of each target gene (Appendix A and Appendix A), and PCR amplifications were performed. SuperReal PreMix Plus (SYBR Green) (TIANGEN Biotech Co., Ltd., Beijing, China) was used to detect the expression levels of the target genes (*kguE*, *kguK*, *kguT*, and *kguD*). The relative expression levels of the genes were calculated using the 2^−ΔΔCT^method and *rpoD* as an internal reference [27].

### 2.9. Statistical Analysis

Three replicates were set for data analysis. The results were presented as mean ± standard deviation. Statistical comparisons were performed using one-way analysis of variance and Duncan’s multiple comparison test. The results were considered significant at *p* < 0.01.

## 3. Results and Discussion

### 3.1. Identification of the Structural Genes of the kgu Operon in Pseudomonas plecoglossicida

A DNA fragment containing *ptxS*, *kguE*, *kguK*, *kguT*, and *kguD* was cloned from the *P. plecoglossicida* JUIM01 genome (Figure 1B). Bioinformatics analysis shows that the *kgu* operon is supposed to be composed of *kguE*, *kguK*, *kguT*, and *kguD* [17]. To confirm this prediction, co-transcription analysis was performed on these genes. The total RNA of *P. plecoglossicida* JUIM01 was extracted to prepare cDNA by reverse transcription PCR. Using P1/P2 and P3/P4 as the primers (Appendix A and Figure 1B) and the cDNA as the template, the *kguE-kguK* and *kguK-kguT* target DNA fragments’ transgene regions were amplified, respectively. Meanwhile, the genome DNA and ddH_2_O were chosen as positive and negative control templates. Agarose gel electrophoresis analysis showed that the sizes of the two target fragments (311 and 400 bp) were consistent with their positive controls (Appendix A). Further sequencing confirmed the results, indicating *kguE*, *kguK*, and *kguT* were co-transcribed. Due to the fact that there is only 9 bp between *kguT* and *kguD* (impossible to contain a promoter), and they have same transcription directions, we concluded *kguT* and *kguD* also transcribed jointly. In summary, *kguE*, *kguK*, *kguT*, and *kguD* were located within the same operon and underwent co-transcription in *P. plecoglossicida*.

We have found a promoter in the upstream of *ptxS* and a terminator in the *ptxS-kgu*E transgene region in *P. plecoglossicida* [17]. In view of the prediction that *ptxS* is independent of the *kgu* operon [17], there should be a promoter in the *ptxS-kgu*E transgene region. The *lacZ* reporter gene fusions were conducted to prove it. Taking K1/K2 as the primers (Appendix A and Figure 1), the *ptxS-kguE* transgene region fragment was amplified from the JUIM01 genome and integrated into the pME6522 vector to obtain the recombinant promoter probe pME6522-*kgu*. The pME6522-*kgu* and empty vector pME6522 were transformed into *P. plecoglossicida* JUIM01 to construct the recombinant strain JUIM01/pME6522-*kgu* and its control JUIM01/pME6522, respectively. Then, the growth and *β*-galactosidase activities of the two strains were compared during cultivation. No significant difference in growth was observed (Appendix A). However, there was a significant difference in their *β*-galactosidase activities (*p* < 0.01). After 12 and 24 h cultivation, the *β*-galactosidase activities of JUIM01/pME6522-*kgu* were 1.21- and 2.16-fold that of JUIM01/pME6522, respectively (Appendix A). The results indicated that there was a promoter between *ptxS* and *kguE*, confirming the prediction that *ptxS* transcribes independently. In conclusion, the *kgu* operon consisted of four structural genes, *kguE*, *kguK*, *kguT*, and *kguD*, and it is supposed to play important roles in the 2KGA catabolism of *P. plecoglossicida* based on our previous studies [5,17,23].

### 3.2. Effects of the Structural Genes of the kgu Operon on 2KGA Catabolism in P. plecoglossicida

We further constructed four knockout strains (named JUIM01Δ*x*, where *x* represents any gene of the *kgu* operon) and their corresponding complementation strains (named JUIM01Δ*x-x*) from *P. plecoglossicida* JUIM01 and comprehensively investigated the 2KGA utilization abilities of the recombinant strains to verify the roles of these genes in 2KGA catabolism. The strains (with JUIM01 as a control) were cultured in seed media containing 2KGA as the main carbon source to observe 2KGA consumption and cell growth. The results showed that the knockout of the genes had apparent impacts on the 2KGA utilization ability of JUIM01. The deletion of *kguK* and *kguE* reduced the average consumption rate of 2KGA (the 2KGA consumed per hour in the period from 0 h to the time 2KGA was used up), delaying the time for 2KGA depletion in the media from 16 to 22 and 30 h, respectively, and the time for the recombinant strains to reach their highest OD_650_ values was also delayed accordingly. After the knockout of *kguT* and *kguD*, the recombinants almost completely lost their abilities to utilize 2KGA (Figure 2A,B). Then, the four structural genes were integrated into the expression vectors and complemented back into the corresponding gene knockout strains, respectively. Afterwards, the 2KGA utilization abilities in the obtained complementation strains were restored (Figure 2C,D). The results indicated that all four structural genes of the *kgu* operon were involved in the 2KGA catabolism of *P. plecoglossicida* and played important roles.

According to existing studies, 2KGA is synthesized in the periplasmic space and then secreted into the extracellular space or enters the cytoplasm for metabolism in *Pseudomonas*. The intracellular metabolic pathway mainly includes the following aspects: 2KGA in the periplasmic space is transported into the cell by the 2KGA transporter KguT and phosphorylated into 2-keto-6-phosphogluconate by the 2-keto-6-phosphogluconate kinase KguK, which is then reduced to 6-phosphogluconate by the 2-keto-6-phosphogluconate reductase KguD, and further enters the Entner–Doudoroff pathway for metabolism (Figure 1A). Interestingly, the importance of KguE, KguK, KguT, and KguD in 2KGA utilization seems to be neither consistent with their order in the 2KGA catabolic pathway nor their physical sequence in the *kgu* operon. The inactivation of KguT possibly stopped the transportation of 2KGA into cell, leading to JUIM01 losing its capability to utilize 2KGA. KguK is supposed to catalyze the first reaction of 2KGA catabolism after 2KGA enters the cytoplasm. However, the 2KGA utilization was not completely inhibited in JUIM01Δ*kguK*, indicating there might be an alternative pathway in the cytoplasm. Although the reaction catalyzed by KguD is after that by KguK, the inactivation of KguD led to greater inhibition in 2KGA utilization, suggesting KguD might have an unknown extra function. KguE is predicted to be an epimerase with an unknown function and has never been annotated in the central metabolic pathway of *Pseudomonas* [11,16]. This study proved for the first time that KguE did participate in 2KGA metabolism in *Pseudomonas*.

During 2KGA fermentation, when the substrate glucose is completely consumed, the produced 2KGA is then utilized by *Pseudomonas* as an alternative carbon source, resulting in reduced production [5]. The knockout of *kguE*, *kguK*, *kguT*, or *kguD* could effectively alleviate or even inhibit the reutilization of 2KGA, indicating that the *kgu* operon would be an ideal target for the modification of 2KGA-producing strains.

### 3.3. 2KGA Fermentation Using the kguK/kguE/kguT/kguD-Knockout and Complementation Recombinants Derived from P. plecoglossicida JUIM01

In industrial production, the reuse of 2KGA by production strains is a problem that is detrimental to the final yield. In the 2KGA utilization experiments, the utilization of 2KGA by JUIM01Δ*kguK/kguE/kguT/kguD* was delayed or even completely blocked. These strains showed potential as industrial production strains for 2KGA. Therefore, we further investigated the 2KGA fermentation capacity of the *kguK/kguE/kguT/kguD*-knockout and complementation recombinants in shake flasks. JUIM01Δ*kguK/kguE/kguT/kguD* exhibited glucose utilization ability similar to JUIM01 (Figure 3A), while the accumulation ability of 2KGA underwent significant changes, as shown in the results of the 2KGA utilization experiments (Figure 3B). Due to the inability of JUIM01Δ*kguT* and JUIM01Δ*kguD* to reuse 2KGA as a growth carbon source, their maximum OD_650_ of 10.4 was significantly lower than the JUIM01 of 16.2. However, JUIM01Δ*kguK* and JUIM01Δ*kguE* took a longer time to reach the maximum OD_650_ of above 15 (Figure 3C). Among them, JUIM01Δ*kguE* might be the best candidate for 2KGA industrial production, as it left a longer time to recover 2KGA from the broth before it was rapidly reused without affecting its cell growth. In the complementation strains, the glucose utilization, synthesis and utilization of 2KGA, and cell growth were similar to those of JUIM01 (Figure 3D–F).

### 3.4. Transcriptional Start Site of the kgu Operon in P. plecoglossicida

The transcription start site of the *kgu* operon in *P. plecoglossicida* was determined by primer extension. First, a primer *kguE*-PE(FAM) (Appendix A and Appendix A) was designed and synthesized for the *kgu* operon and its upstream sequence, and then hybridized with the RNA of JUIM01. The result showed two sharp peaks at around 120 bp from the 5′ end of the specific reverse primer, corresponding to the base pair TT/AA, which was identified as the transcription start site of the *kgu* operon (Figure 4). This site was located 34 bp upstream of the initiation codon of *kguE* (the first structural gene of the *kgu* operon). It is worth noting that the transcription start site was located within a 14 bp palindrome (5′-TGA**AA**CCGGTTTCA-3′, the letters in bold represent the transcription start site).

### 3.5. Specific Binding Between PtxS and the kgu Operon Promoter Region in P. plecoglossicida

In *P. aeruginosa* and *P. putida*, the transcription regulator PtxS specifically recognizes and binds to the 14 bp palindrome (5′-TGAAACCGGTTTCA-3′) in the *kgu* operon promoter region, thereby regulating the transcription of the *kgu* operon [11,16]. A previous study has shown that PtxS from *P. plecoglossicida* can also specifically bind to the 14 bp palindrome [22]. Therefore, PtxS is likely to regulate the transcription of the *kgu* operon and 2KGA metabolism by specifically binding to the region where the 14 bp palindrome is located in *P. plecoglossicida*. To verify this hypothesis, based on the *kgu* operon and its upstream sequence, a specific probe JUIM01-*kgu*(FAM) labeled with FAM was synthesized. PtxS from JUIM01 was then heterologously expressed and purified. Finally, the interaction between PtxS and the *kgu* operon promoter region was investigated by EMSA. Reactions were implemented using a 50-ng labeled probe, and without or with 0.5 μg of PtxS. As shown in Figure 5, in the absence of PtxS, the band of the free DNA probe is displayed at the bottom of the gel (Lane 1). As PtxS was added, the molecular weight of the protein–probe complex became larger, and its mobility was restrained correspondingly (Lane 2). The results confirmed that PtxS could specifically bind to the *kgu* operon promoter region of *P. plecoglossicida*.

In *P. aeruginosa* and *P. putida*, the regulator PtxS regulates the metabolism of 2KGA, which acts as an effector of PtxS. 2KGA can bind to PtxS to release the binding between PtxS and the regulated DNA, thereby relieving the repression of the corresponding genes at the transcriptional level [11,16]. In *P. plecoglossicida*, 2KGA can also bind to PtxS [22]. To clarify the effect of 2KGA on the specific binding of PtxS to the *kgu* operon promoter region in *P. plecoglossicida*, different concentrations of 2KGA were added to the EMSA systems (Figure 5). When 2KGA was not present, PtxS bound to JUIM01-*kgu*(FAM), and the band was located at the top of the gel (Lane 2). With the addition of 2KGA, the bands showed accelerated mobility and shifted to lower positions (Lane 3–5), indicating that the retardation of mobility caused by the binding of PtxS–probe was relieved. In summary, similar to *P. aeruginosa* and *P. putida*, in *P. plecoglossicida*, PtxS could also specifically bind to the *kgu* operon promoter region, while 2KGA acted as an effector binding to PtxS, thereby releasing PtxS from the *kgu* operon promoter region.

DNaseI footprint analysis further revealed the binding site of PtxS in the *kgu* operon promoter region. Compared with the control group without PtxS, a distinct protective region appeared when 0.9 and 2.2 μg PtxS was added to the systems with 400 ng of JUIM01-*kgu*(FAM). The protective region covered a sequence of about 26 bp (5′-AATGA**AA**CCGGTTTCATCTGACAAGA-3′) (Figure 6). Interestingly, the 14 bp palindrome (the underlined bases) was found in this region, located 25–39 bp upstream of *kguE*.

### 3.6. Transcriptional Regulation of PtxS on the kgu Operon in P. plecoglossicida

To explore the transcriptional regulation mechanism of PtxS on the *kgu* operon in *P. plecoglossicida*, a *ptxS* gene-knockout strain JUIM01Δ*ptxS* was constructed. The expression levels of each gene within the *kgu* operon before and after *ptxS* knockout were compared by RT-qPCR. Under the same culture conditions, the expression levels of the structural genes in the *kgu* operon in JUIM01Δ*ptxS* were significantly up-regulated to 1.75–2.39-fold compared to JUIM01 (*p* < 0.01) (Appendix A), indicating PtxS negatively regulated the expression of the *kgu* operon.

The negative regulation of PtxS was further verified by *lacZ* reporter gene fusions. The recombinant plasmid pME6522-*kgu* was transformed into JUIM01 and JUIM01Δ*ptxS* to obtain the recombinant strains JUIM01/pME6522-*kg*u and JUIM01Δ*ptxS*/pME6522-*kgu*, respectively. The results showed no significant difference in the growth between the two strains (Appendix A), but a significant difference in their *β*-galactosidase activities (*p* < 0.01): after 12 and 24 h cultivation, JUIM01Δ*ptxS*/pME6522-*kgu* was 1.87- and 1.37-fold of JUIM01/pME6522-*kgu*, respectively (Appendix A), indicating the presence of *ptxS* had a repressive effect on the fusion expression of *kgu-lacZ*, thereby demonstrating that PtxS negatively regulated the expression of the *kgu* operon at transcriptional level.

In natural edatope, 2KGA is one of the most commonly secreted organic acids by various rhizosphere bacteria including *Pseudomonas*, which is beneficial to plant growth and root colonization [28,29,30,31,32]. The repression of PtxS on the *kgu* operon transcription provides a mechanism for 2KGA secretion in *Pseudomonas*. Additionally, during fermentation, 2KGA is both the target product and an intermediate metabolite that can be utilized by *Pseudomonas* [9,33,34,35,36,37]. The negative regulation of PtxS on the *kgu* operon is essential to 2KGA accumulation.

## 4. Conclusions

In this study, the structural genes of the *kgu* operon in *P. plecoglossicida* were first identified. Unlike *P. aeruginos*a reported by Swanson et al. (2000) [16], the *kgu* operon in *P. plecoglossicida* consisted of four structural genes, *kguE*, *kguK*, *kguT*, and *kguD*. The expression of *ptxS* located upstream of *kguE* is independent of the operon. Subsequently, the role of each structural gene in 2KGA metabolism was determined via gene knockout and complementation, demonstrating that the *kgu* operon was involved in and essential to 2KGA metabolism in *P. plecoglossicida*. The results indicated the deletion of gene(s) in the *kgu* operon would be an ideal strategy to improve 2KGA accumulation. The knockout stains, especially JUIM01Δ*kguE*, showed potential as industrial production strains for 2KGA. In this study, we proved that KguE participated in glucose/2KGA metabolism in *Pseudomonas*, which was barely mentioned in related studies. However, the reaction KguE exactly catalyzes remains to be studied. The 2KGA utilization was not completely inhibited in the *kguK*-knockout mutant, indicating there might be an alternative pathway for 2KGA catabolism in the cytoplasm. At last, the regulation of PtxS on the *kgu* operon was analyzed using multiple methods. In *P. plecoglossicida*, the LacI-family transcriptional regulator PtxS recognized the 14 bp palindrome located in the *kgu* operon promoter region and specifically bound to a 26 bp region containing the palindrome (Figure 7A). Since the binding site overlapped with the transcription start site, the binding of PtxS possibly hindered the binding of RNA polymerase, thereby inhibiting the transcription of the *kgu* operon and further regulating 2KGA metabolism. Meanwhile, 2KGA could bind to PtxS as an effector, dissociating from the *kgu* operon promoter region, thereby releasing the transcriptional repression (Figure 7B). The above transcriptional regulation mechanism was similar to that reported in *P. aeruginosa* and *P. putida* and might be a common mechanism in the genus *Pseudomonas*.

## Figures and Tables

**Figure 1 microorganisms-12-02530-f001:**
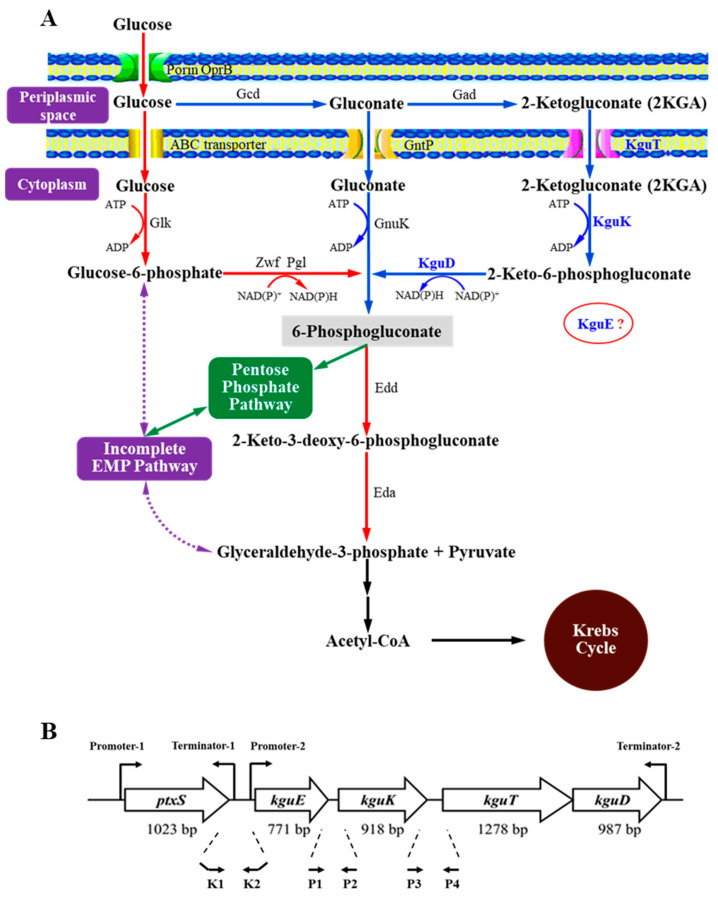
(**A**) The deduced glucose metabolism in *Pseudomonas* based on published studies, and (**B**) physical map of *ptxS* and the putative *kgu* operon (*kguE*, *kguK*, *kguT*, and *kguD*) in *P. plecoglossicida* JUIM01. Gcd, glucose dehydrogenase; Gad, gluconate dehydrogenase (encoded by 3 genes: *gndS*, *gndL*, and *gndC*); Glk, glucokinase; Zwf, glucose-6-phosphate dehydrogenase; Pgl, 6-phosphogluconolactonase; GntP, gluconate permease; GnuK, gluconokinase; KguT, 2-ketogluconate transporter; KguK, 2-ketogluconate kinase; KguD, 2-keto-6-phosphogluconate reductase; KguE, a putative isomerase with unknown function; Edd, 6-phosphogluconate dehydratase; Eda, 2-keto-3-deoxy-6-phosphogluconate aldolase. K1, K2, P1, P2, P3, and P4 are primers used for the PCR amplification of the intergenic regions.

**Figure 2 microorganisms-12-02530-f002:**
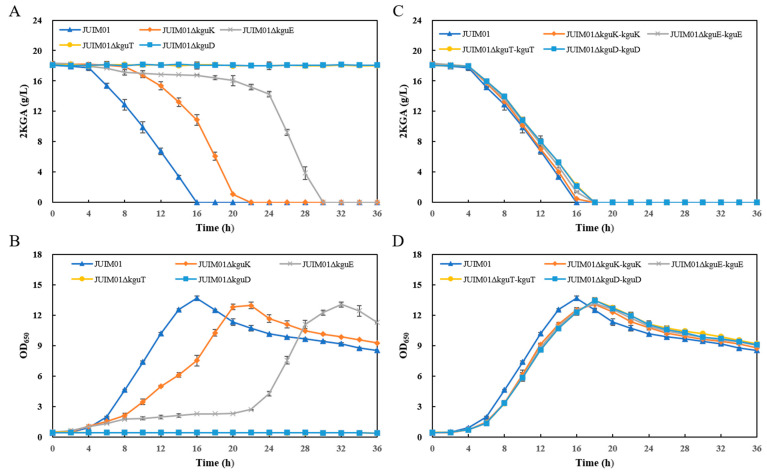
2KGA utilization experiments of the (**A**,**B**) *kguK/kguE/kguT/kguD*-knockout and (**C**,**D**) complementation recombinants.

**Figure 3 microorganisms-12-02530-f003:**
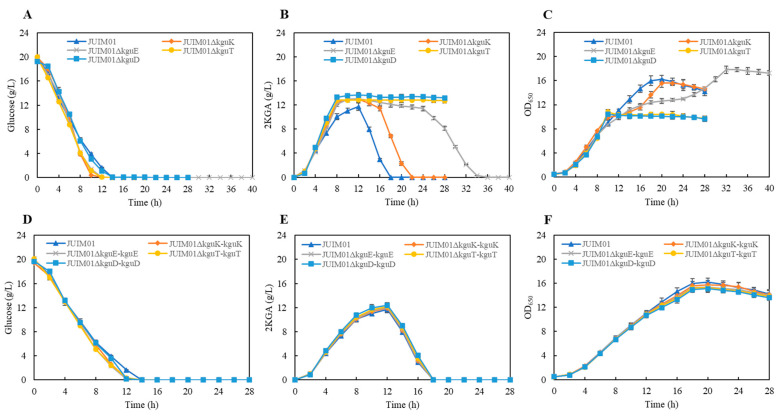
2KGA fermentation of the (**A**–**C**) *kguK/kguE/kguT/kguD*-knockout and (**D**–**F**) complementation recombinants.

**Figure 4 microorganisms-12-02530-f004:**
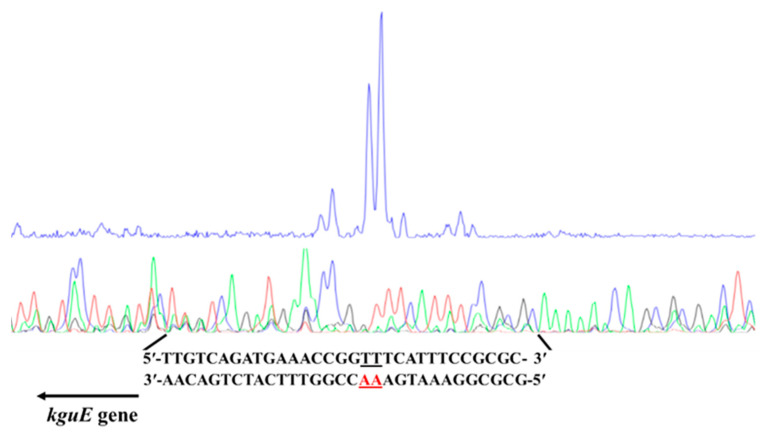
Determination of the transcriptional start site of the *kgu* operon from *P. plecoglossicida* by primer extension analysis. The AA letters in red represent the transcription start site.

**Figure 5 microorganisms-12-02530-f005:**
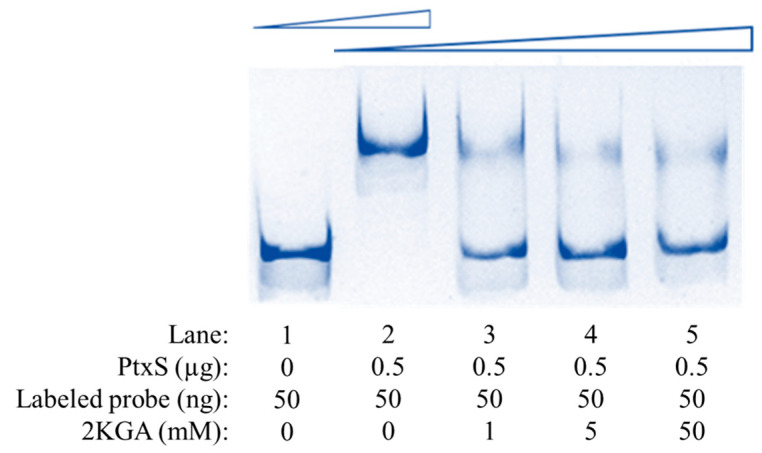
EMSA analysis of the binding between PtxS and the *kgu* operon from *P. plecoglossicida* (Lane 1 and 2) and the effect of 2KGA on the PtxS-*kgu* operon binding (Lane 2–5).

**Figure 6 microorganisms-12-02530-f006:**
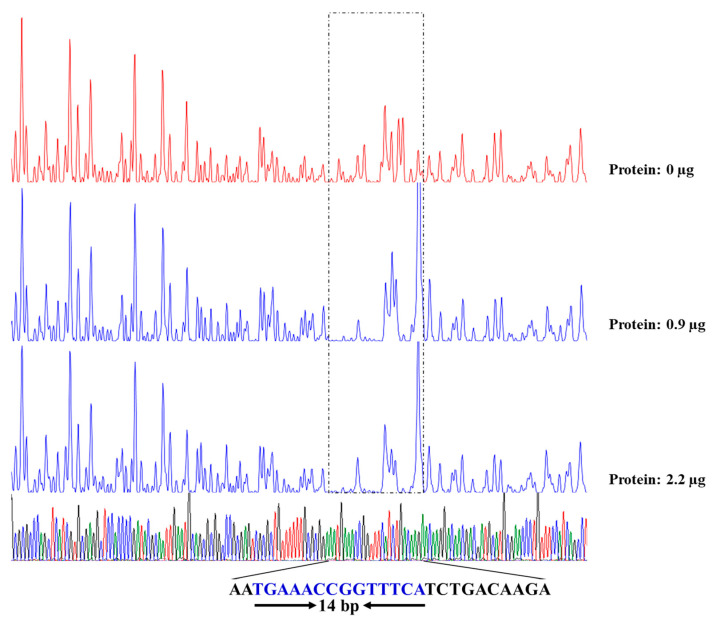
Determination of the binding sites of the *kgu* operon promoter region from *P. plecoglossicida* with PtxS by DNase I footprinting analysis. The letters in blue represent a 14 bp palindrome within the 26 bp binding site.

**Figure 7 microorganisms-12-02530-f007:**
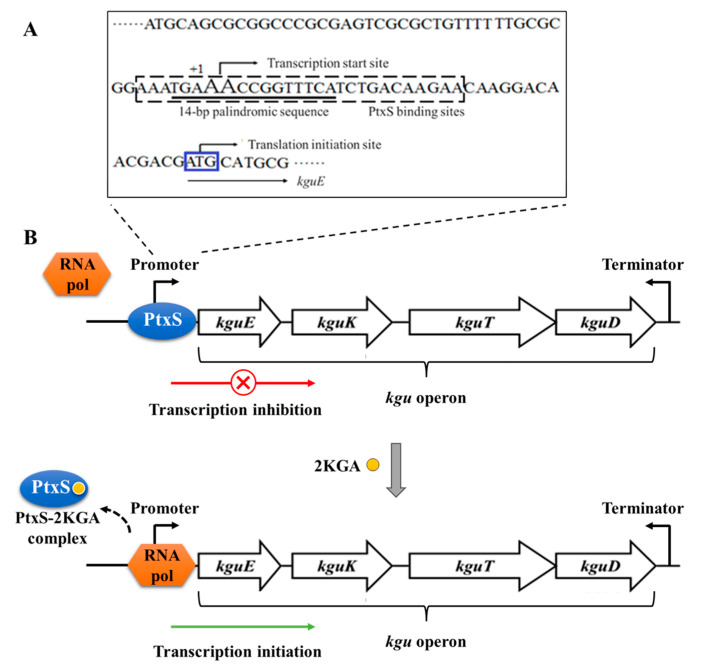
(**A**) Analysis of the *kgu* operon promoter region in *P. plecoglossicida* and (**B**) the transcriptional regulation mechanism of PtxS on the *kgu* operon in *P. plecoglossicida*. The AA with larger font size indicates the transcription start site, the sequence in the dotted box indicates the PtxS binding sites, the underlined sequence is the 14 bp palindrome, and the ATG in the blue box is the start codon of the *kguE* gene (the first ORF in the *kgu* operon).

**Table 1 microorganisms-12-02530-t001:** Strains and plasmids.

Strains and Plasmids	Description	Sources
**Strains**		
*P. plecoglossicida* JUIM01	Industrial 2KGA-producing strain	Our lab [4,5,6,7,8]
JUIM01Δ*kguE*	Derived from JUIM01 with *kguE* knockout	This work
JUIM01Δ*kguK*	Derived from JUIM01 with *kguK* knockout	Our lab [23]
JUIM01Δ*kguT*	Derived from JUIM01 with *kguT* knockout	Our lab [5]
JUIM01Δ*kguD*	Derived from JUIM01 with *kguD* knockout	This work
JUIM01Δ*ptxS*	Derived from JUIM01 with *ptxS* knockout	This work
JUIM01Δ*kguE*-*kguE*	Derived from JUIM01Δ*kguE* with *kguE* complemented	This work
JUIM01Δ*kguK*-*kguK*	Derived from JUIM01Δ*kguK* with *kguK* complemented	Our lab [23]
JUIM01Δ*kguT*-*kguT*	Derived from JUIM01Δ*kguT* with *kguT* complemented	Our lab [5]
JUIM01Δ*kguD*-*kguD*	Derived from JUIM01Δ*kguD* with *kguD* complemented	This work
JUIM01/pME6522	Negative control for *lacZ* fusion analysis, Tc^R^	This work
JUIM01/pME6522-*kgu*	Recombinant strain for *lacZ* fusion analysis, Tc^R^	This work
JUIM01Δ*ptxS*/pME6522-*kgu*	Recombinant strain for *lacZ* fusion analysis, Tc^R^	This work
*Escherichia coli* BL21(DE3)/pET-28a-*ptxS*	PtxS heterologous-expression strain, Kan^R^	Our lab [22]
**Plasmids**		
pME6522	*E. coli*-*Pseudomonas* shuttle vector for transcriptional *lacZ* fusions and promoter probing, Tc^R^	[24]
pME6522-*kgu*	Derived from pME6522, containing a 132 bp *ptxS*-*kguE* intergenic sequence, Tc^R^	This work
pK18*mobSacB*	Suicide vector for in-frame deletions, Kan^R^	Our lab [23]
pK18*mobSacB*-Δ*kguE*	Derived from pK18*mobSacB*, containing an incomplete *kguE* gene, for in-frame deletion of *kguE* in JUIM01, Kan^R^	This work
pK18*mobSacB*-Δ*kguD*	Derived from pK18*mobSacB*, containing an incomplete *kguD* gene, for in-frame deletion of *kguD* in JUIM01, Kan^R^	This work
pK18*mobSacB*-Δ*ptxS*	Derived from pK18*mobSacB*, containing an incomplete *ptxS* gene, for in-frame deletion of *ptxS* in JUIM01, Kan^R^	This work
pBBR1MCS-2	*E. coli*-*Pseudomonas* shuttle vector for expression, Kan^R^	Our lab [23]
pBBR*kguE*	Derived from pBBR1MCS-2, for constitutive expression of *kguE*, Kan^R^	This work
pBBR*kguD*	Derived from pBBR1MCS-2, for constitutive expression of *kguD*, Kan^R^	This work

## Data Availability

The original contributions presented in the study are included in the article/Supplementary Material, further inquiries can be directed to the corresponding author.

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
