# Peer review of "Characterization and Transcriptional Regulation of the 2-Ketogluconate Utilization Operon in Pseudomonas plecoglossicida"

_microorganisms, 2024, doi:10.3390/microorganisms12122530_

Round 1
Reviewer 1 Report
Comments and Suggestions for Authors
The authors of the paper “Characterization and transcriptional regulation of the 2-ketogluconate utilization operon in Pseudomonas plecoglossicida” performed a complete transcriptional analysis. However, their main conclusions are the same as those already reached in different studies (sometimes with other organisms, though) regarding the kgu operon. Therefore, although the work is coherent, is based on its own results and the experimentation is correct and well designed, the originality of the work is not very high. Perhaps the authors should indicate, more forcefully, what makes the work novel. Beyond that, I would have to make other comments. For example.
1. When in section 3.1. The primers used are being described, it would be very useful if they were drawn on Figure 1.
2. The quality/resolution of Figure 2 would have to be increased.
3. Why do the authors believe that “only” the disappearance of 2KGA is delayed in the KguK and KguE mutants? How is 2KGA ultimately metabolized?
4. Although in the first paragraph on page 7 the authors review the functions of the different gene products of the Kgu cluster, the possible role of KguE remains unknown. To that point, it is known that its absence causes a delay in the assimilation of 2KGA, which is also accompanied by a delay in growth, but only that. Have you performed sequence homology analysis to try to elucidate its role in the catabolic process? Could the degradation route of 2KGA be hypothesized including all the structural components of the cluster?
Author Response
Comment 0: The authors of the paper “Characterization and transcriptional regulation of the 2-ketogluconate utilization operon in Pseudomonas plecoglossicida” performed a complete transcriptional analysis. However, their main conclusions are the same as those already reached in different studies (sometimes with other organisms, though) regarding the kgu operon. Therefore, although the work is coherent, is based on its own results and the experimentation is correct and well designed, the originality of the work is not very high. Perhaps the authors should indicate, more forcefully, what makes the work novel.
Response 0: Thank you for your suggestion. Revision has been made to emphasize the novelty of the manuscript. Especially, 2KGA fermentation data using the mutants have been added in Section 3.3.
Comment 1: When in section 3.1. The primers used are being described, it would be very useful if they were drawn on Figure 1.
Response 1: Thank you for your suggestion. The primers have been added on Figure 1B.
Comment 2: The quality/resolution of Figure 2 would have to be increased.
Response 2: Yes, the resolution of Figure 2 has been improved. Please check it in the Word file, as the resolution may decrease in the PDF file.
Comment 3: Why do the authors believe that “only” the disappearance of 2KGA is delayed in the KguK and KguE mutants? How is 2KGA ultimately metabolized?
Response 3: (1) The fermentation data showed that there were no significant differences in glucose consumption or 2KGA synthesis, but in 2KGA catabolism between JUIM01 and the mutants. In order to clearly present the differences, the data on 2KGA fermentation has been added.
(2) A figure on glucose and 2KGA metabolism in Pseudomonas has also been added in the revised manuscript (Figure 1A). According to published reports so far, in wild-type Pseudomonas, 2KGA in the periplasmic space is transported into the cell by KguT, and phosphorylated into 2-keto-6-phosphogluconate by KguK, which is then reduced to 6-phosphogluconate by KguD, and further enters the Entner-Doudoroff pathway for metabolism, while KguE is not annotated in the pathway. In this study, the knockout of kguE showed an adverse effect on 2KGA catabolism, indicating KguE was involved in 2KGA catabolism, but its function remained unknown. Interestingly, based on the published studies, the knockout of kguK is supposed to block the 2KGA catabolic pathway, just like the knockout of kguT and kguD. However, the deletion of kguK did not block 2KGA catabolism, but reduced the 2KGA consumption rate, delaying the time for 2KGA depletion in the media, indicating there might be an alternative pathway for 2KGA catabolism in the cytoplasm. Our subsequent research has also confirmed this hypothesis (data not published).
Comment 4: Although in the first paragraph on page 7 the authors review the functions of the different gene products of the Kgu cluster, the possible role of KguE remains unknown. To that point, it is known that its absence causes a delay in the assimilation of 2KGA, which is also accompanied by a delay in growth, but only that. Have you performed sequence homology analysis to try to elucidate its role in the catabolic process? Could the degradation route of 2KGA be hypothesized including all the structural components of the cluster?
Response 4: Yes, we did perform sequence homology analysis. The BLAST results showed that KguE from JUIM01 has 99.61% identity with a sugar phosphate isomerase/epimerase from Pseudomonas putida (Accession: WP_084858275.1), and 99.22% identity with an AP endonuclease from Pseudomonas putida (Accession: WP_324813500.1). However, there are no isomers among the metabolites in reported 2KGA metabolic pathway (Figure 1A), and AP endonuclease does not show a direct correlation with 2KGA metabolism. Furthermore, we were not able to find studies on these two enzymes. Therefore, it is hard to hypothesize the function of KguE in 2KGA metabolism by sequence homology analysis. We will appreciate it if you have any suggestions on it.
Reviewer 2 Report
Comments and Suggestions for Authors
In their manuscript, Sun et al. investigated the organization and regulation of the 2-ketogluconate utilization operon in P. plecoglossicida. Since 2KGA is used as a precursor for production of an antioxidant widely used for food production the improvement of 2KGA production is of principal interest. The research is well conducted in most cases, however, there are some open points that should be addressed to improve the quality of the manuscript before it is suitable for publication.
Major points:
Figure 1: The position and direction of the used oligos for PCR, RT-PCR and Primer extension should be indicated in the figure. Moreover, the predicted positions of promoters and transcription terminators should also be included.
Line 175: I totally agree that there is a promoter in the intergenic region between ptxS and kguE. However, just because there is a promoter did not exclude that ptxS is also part of the operon since it is not excluded that the proposed promoter is an alternative promoter. At least a transcription terminator has to be predicted between ptxS and kguE. If this is not possible a PCR with primers binding to ptxS and kguE has to be conducted according to the experiment in Fig. S1.
Line 184ff: The effect after 12 h is very weak. How often was this experiment repeated (technical and biological replicates) to justify significance. Moreover, the growth of the strains is very poor whereby even small variations in sampling could lead to large variations in the reporter gene assay.
Figure 2/S2: Why is there a different OD (OD600 vs OD650) used for OD measurement? Why is the OD of the cultures used for the lacZ assay so much lower compared to the OD for the KGA consumption assay (0.3 vs 10 after 12h) Please explain.
Figure S4: The bars in Fig. S4 look exactly like the bars in Fig. S2. However, the values for the fold-changes given in the manuscript are different. Please explain.
Line 203: I don’t entirely agree that the KGA consumption rate is reduced upon deletion of the kgu genes. While the WT starts consumption after 4h and is finished within 12 h, the kguK deletion strain starts with consumption after 8 h and is nearly finished after 12 h which is more or less the same consumption rate, however it is delayed. In contrast, there are clearly two phases for KGA consumption in the kguE deletion strain. The 1st phase from 4h-24h where nearly no KGA is consumed (or KGA is consumed with a strongly reduced consumption rate) and a second phase from 24h to 30h were KGA is very fast consumed (with a consumption rate nearly twice as high as the WT). Please explain.
Fig. 4A: The quality of the EMSA is not satisfactory. Obviously, by addition of 0.9 µg PtsX all DNA is already shifted. Consequently, addition of more PtsX results either in super-shifts (if additional PtsX binding sites are used) or more likely in unspecific binding of PtsX to the fragment. The EMSAs should be repeated with concentrations between 0 and 0.9 µg protein resulting in partially shifted DNA (ideally including the 1:1 ratio). Moreover, the dissociation constant (KD) for the interaction should be calculated. All subsequent experiments should also be conducted with the lowest PtsX concentration suitable to result in a complete shift. The position of the band in lane 5 is not on the same height as the band in lane 1. Together with the lack of partially shifted DNA this opens the question if the shift in lane 2 is already a supershift and the real shift is on the height of the band in lane 5. This issue will easily clarify when repeating the shifts with lower protein concentrations. Moreover, to prove specificity of the interaction shifts with non-specific competitors in 100, 1000 or even 10000 fold molar excess could be conducted.
Fig. 4B: The gel shown is totally different compared to the gel in 4A (running distance, size, position of the shifts). Thus, it is indispensable to include a control lane without protein there. Moreover, 1 mM KGA seems to be enough to revert the shift while an increase to 5 or 50 mM has no further effect. Consequently, concentrations between 0 and 1 mM KGA would be more interesting.
Conclusion: The finding that the kgu-operon is a suitable target to improve KGA production is the most important finding of the manuscript. Thus, please consider to move the discussion part dealing with this finding towards the end of the discussion section.
Figure S6: The model for the function of the operon is lost if it is hidden in the supplements. It should be included in the manuscript. However, the model should be improved. It is not necessary to draw the DNA as helical ribbon. A schematic drawing is enough and allows to include important information (relative positions of operator, promoter elements, transcription start site, ORF). The middle picture is not correct, as your experiments showed that PtsX did not bind to DNA in the presence of KGA, and can be removed. The RNAP should also be included in the upper picture and the lack of binding of RNAP indicated.
Figure
Minor points:
Line 12: The reporter gene fusion is suitable to determine the strength of a putative promoter sequence while RT-PCR is suitable to identify a transcript. Thus, the phrasing that structural genes were identified with the two methods is somewhat unfortunate and should be reworked.
Line 14: Please rephrase this sentence.
Line 18: please use a capital “L” for LacI-family since this refers to the protein LacI.
Line 25: What do you mean with “theoretical bases”? Prerequisite? Please explain.
Line 161: Do you mean: “… and cDNA prepared by RT-PCR.”?
Line 189-192: This was already stated in the introduction and can be removed.
Line 192: Please remove “preliminary”.
Line 200: Please remove “above”
Line 203: Please remove “specifically”
Line 219-221: It is not necessary to put the protein names in brackets.
Line 242: Either use “is the transcription start”, “must be the transcription start” or “was identified as the transcription start”.
Line 265: Free DNA did not shift. A shift only occurs if the labelled probe interacts with its interaction partner to form a complex and thus migrates slower. Please correct.
Line 271: Please remove “that is” and make two sentences instead.
Line 273: Please use “repression … is relieved” instead of “disinhibition”
Line 302: An upregulation means that the effect is > 1-fold. A 0.75-fold effect would be a down-regulation. Please correct.
Line 323: Expression of ptsX is independent from kguE operon. Please correct.
Line 326: The kgu-operon is only essential for KGA catabolism.
Line 333: Please use “binds” instead of “bound”.
Author Response
Major points:
Comment 1: Figure 1: The position and direction of the used oligos for PCR, RT-PCR and Primer extension should be indicated in the figure. Moreover, the predicted positions of promoters and transcription terminators should also be included.
Response 1: Yes, the predicted positions of promoters and transcription terminators and the primers K1, K2, P1, P2, P3, and P4 have been added on Figure 1B. The position and direction of other primers have been added in the revised supplementary materials (Figure S1), as there were many primers used in the study.
Comment 2: Line 175: I totally agree that there is a promoter in the intergenic region between ptxS and kguE. However, just because there is a promoter did not exclude that ptxS is also part of the operon since it is not excluded that the proposed promoter is an alternative promoter. At least a transcription terminator has to be predicted between ptxS and kguE. If this is not possible a PCR with primers binding to ptxS and kguE has to be conducted according to the experiment in Fig. S1.
Response 2: Yes, a terminator was found between ptxS and kguE in our previous study (Luan et al., 2018). The sentence “We have found a promoter in the upstream of ptxS and a terminator in the ptxS-kguE transgene region in P. plecoglossicida” has been added in Section 3.1.
Comment 3: Line 184ff: The effect after 12 h is very weak. How often was this experiment repeated (technical and biological replicates) to justify significance. Moreover, the growth of the strains is very poor whereby even small variations in sampling could lead to large variations in the reporter gene assay.
Response 3: Three replicates were set for data analysis. Both the strains were cultured under the same conditions.
Comment 4: Figure 2/S2: Why is there a different OD (OD600 vs OD650) used for OD measurement? Why is the OD of the cultures used for the lacZ assay so much lower compared to the OD for the KGA consumption assay (0.3 vs 10 after 12h) Please explain.
Response 4: (1) Thank you for pointing this out. It should be OD650 on Figure S2. The mistake has been revised.
(2) The OD of the cultures used for the lacZ assay was different from the OD for the 2KGA consumption assay, because the stains were cultured under very different conditions. In the lacZ assay, the strains were cultured in 5 mL LB liquid media in tubes, while in the 2KGA consumption assay, the strains were cultured in 50 mL seed media in 500-mL shake flasks. The difference in conditions led to difference in growth.
Comment 5: Figure S4: The bars in Fig. S4 look exactly like the bars in Fig. S2. However, the values for the fold-changes given in the manuscript are different. Please explain.
Response 5: Sorry, the same figure was used by mistake. The correct figure has been attached in the revised supplementary materials. Thank you for pointing this out.
Comment 6: Line 203: I don’t entirely agree that the KGA consumption rate is reduced upon deletion of the kgu genes. While the WT starts consumption after 4h and is finished within 12 h, the kguK deletion strain starts with consumption after 8 h and is nearly finished after 12 h which is more or less the same consumption rate, however it is delayed. In contrast, there are clearly two phases for KGA consumption in the kguE deletion strain. The 1st phase from 4h-24h where nearly no KGA is consumed (or KGA is consumed with a strongly reduced consumption rate) and a second phase from 24h to 30h were KGA is very fast consumed (with a consumption rate nearly twice as high as the WT). Please explain.
Response 6: (1) The “2KGA consumption rate” mentioned here means the average 2KGA consumption rate, that is, the 2KGA consumed per hour (g/L/h) in the period from 0 h to the time 2KGA was used up. As shown in Figure 2, the time for 2KGA consumption of JUIM01, JUIM01ΔkguK, and JUIM01ΔkguE were 16, 22 and 30 h, respectively. Therefore, we described it as “the deletion of kguK and kguE reduced the consumption rate of 2KGA”. The explanation has been added in the manuscript to avoid the ambiguousness.
(2) Yes, we agree that there seems to be two phases for 2KGA consumption in the kguE and kguK deletion strains. However, the mechanism of the phenomenon remains unknown to us.
Comment 7: Fig. 4A: The quality of the EMSA is not satisfactory. Obviously, by addition of 0.9 µg PtsX all DNA is already shifted. Consequently, addition of more PtsX results either in super-shifts (if additional PtsX binding sites are used) or more likely in unspecific binding of PtsX to the fragment. The EMSAs should be repeated with concentrations between 0 and 0.9 µg protein resulting in partially shifted DNA (ideally including the 1:1 ratio). Moreover, the dissociation constant (KD) for the interaction should be calculated. All subsequent experiments should also be conducted with the lowest PtsX concentration suitable to result in a complete shift. The position of the band in lane 5 is not on the same height as the band in lane 1. Together with the lack of partially shifted DNA this opens the question if the shift in lane 2 is already a supershift and the real shift is on the height of the band in lane 5. This issue will easily clarify when repeating the shifts with lower protein concentrations. Moreover, to prove specificity of the interaction shifts with non-specific competitors in 100, 1000 or even 10000 fold molar excess could be conducted.
Comment 8: Fig. 4B: The gel shown is totally different compared to the gel in 4A (running distance, size, position of the shifts). Thus, it is indispensable to include a control lane without protein there. Moreover, 1 mM KGA seems to be enough to revert the shift while an increase to 5 or 50 mM has no further effect. Consequently, concentrations between 0 and 1 mM KGA would be more interesting.
Response 7&8: Thank you for your suggestion, we agree. We have uploaded a new figure of EMSA. We believe that the results can demonstrate the binding of PtxS-DNA, and the effect of 2KGA as an effector. We also agree that concentrations between 0 and 1 mM 2KGA would be more interesting. Unfortunately, due to limited time for revision, we are unable to make further improvement, but we will carefully consider your suggestions in future research.
Comment 9: Conclusion: The finding that the kgu-operon is a suitable target to improve KGA production is the most important finding of the manuscript. Thus, please consider to move the discussion part dealing with this finding towards the end of the discussion section.
Response 9: Yes, discussion on it has been added. Moreover, 2KGA fermentation data using the mutants have been added in Section 3.3.
Comment 10: Figure S6: The model for the function of the operon is lost if it is hidden in the supplements. It should be included in the manuscript. However, the model should be improved. It is not necessary to draw the DNA as helical ribbon. A schematic drawing is enough and allows to include important information (relative positions of operator, promoter elements, transcription start site, ORF). The middle picture is not correct, as your experiments showed that PtsX did not bind to DNA in the presence of KGA, and can be removed. The RNAP should also be included in the upper picture and the lack of binding of RNAP indicated.
Response 10: Yes, the figure has been revised and shown in the manuscript as your suggestion (Figure 7). Thank you.
Minor points:
Comment 1: Line 12: The reporter gene fusion is suitable to determine the strength of a putative promoter sequence while RT-PCR is suitable to identify a transcript. Thus, the phrasing that structural genes were identified with the two methods is somewhat unfortunate and should be reworked.
Response 1: Yes, the sentence has been revised as “In this study, the structural genes of the kgu operon and its promoter in P. plecoglossicida JUIM01 were identified using reverse transcription PCR and lacZ reporter gene fusion.”
Comment 2: Line 14: Please rephrase this sentence.
Response 2: Yes, the sentence has been revised as “The results showed the kgu operon in P. plecoglossicida was composed of four structural genes: kguE, kguK, kguT, and kguD. The ptxS gene locating at the upstream of kguE was excluded from the kgu operon.”
Comment 3: Line 18: please use a capital “L” for LacI-family since this refers to the protein LacI.
Response 3: Yes, the revision has been made.
Comment 4: Line 25: What do you mean with “theoretical bases”? Prerequisite? Please explain.
Response 4: Yes, in our opinion, the construction of a chassis should base on deep understanding of its genome and metabolism, especially the central carbon metabolism. In Pseudomonas species, 2KGA metabolism is an important part of glucose metabolism. Hopefully, our study can provide guidance on subsequent modification of the Pseudomonas strains.
Comment 5: Line 161: Do you mean: “… and cDNA prepared by RT-PCR.”?
Response 5: Yes, the sentence has been revised as “The total RNA of P. plecoglossicida JUIM01 was extracted to prepare cDNA by reverse transcription PCR”.
Comment 6: Line 189-192: This was already stated in the introduction and can be removed.
Response 6: Yes, the sentences have been removed.
Comment 7: Line 192: Please remove “preliminary”.
Response 7: Yes, the word has been removed.
Comment 8: Line 200: Please remove “above”
Response 8: Yes, the word has been removed.
Comment 9: Line 203: Please remove “specifically”
Response 9: Yes, the word has been removed.
Comment 10: Line 219-221: It is not necessary to put the protein names in brackets.
Response 10: Excuse us, do you mean to remove the protein names together with the brackets, or just to remove the brackets, please? We have removed the brackets. Please let us know if the protein names are required to be removed.
Comment 11: Line 242: Either use “is the transcription start”, “must be the transcription start” or “was identified as the transcription start”.
Response 11: Yes, the sentence has been revised as “was identified as the transcription start”.
Comment 12: Line 265: Free DNA did not shift. A shift only occurs if the labelled probe interacts with its interaction partner to form a complex and thus migrates slower. Please correct.
Response 12: Yes, the sentence has been revised as “the band of the free DNA probe was displayed at the bottom of the gel”.
Comment 13: Line 271: Please remove “that is” and make two sentences instead.
Response 13: Yes, the sentence has been revised as your suggestion.
Comment 14: Line 273: Please use “repression … is relieved” instead of “disinhibition”
Response 14: Yes, the sentence has been revised as your suggestion.
Comment 15: Line 302: An upregulation means that the effect is > 1-fold. A 0.75-fold effect would be a down-regulation. Please correct.
Response 15: We meant that the expression levels of the genes was increased by 0.75-1.39 folds (or increased to 1.75-2.39 folds). Therefore, the sentence has been revised as “the expression levels of the structural genes in the kgu operon in JUIM01ΔptxS were significantly up-regulated to 1.75-2.39 folds compared to JUIM01”.
Comment 16: Line 323: Expression of ptsX is independent from kguE operon. Please correct.
Response 16: Yes, the sentence has been revised as your suggestion.
Comment 17: Line 326: The kgu-operon is only essential for KGA catabolism.
Response 17: As shown in the Figure 1A in the revised manuscript, the 2KGA metabolism is an important part of glucose metabolism in Pseudomonas species. Especially, over 80-90% glucose can be converted into 2KGA in P. plecoglossicida JUIM01. Therefore, we believe the kgu operon is important to both the glucose and 2KGA metabolism in the strain.
Comment 18: Line 333: Please use “binds” instead of “bound”.
Response 16: Yes, the word has been revised.
Round 2
Reviewer 1 Report
Comments and Suggestions for Authors
The authors responded appropriately to all previous comments, so, in my opinion, the work is ready for publication.